# The effect of dietary interventions on inflammatory biomarkers among people with multiple sclerosis: A protocol for systematic review and meta-analysis of randomized controlled trials

**Farnoosh Shemirani** [1‡*], **Wade R. Pingel** [2‡], **Tyler J. Titcomb** [1], **Asma Salari-Moghaddam** [3], **Farshad Arsalandeh** [4], **Solange M. Saxby** [1,5], **Linda G. Snetselaar** [6‡], **Terry L. Wahls** [1‡*]

1 Department of Internal Medicine, University of Iowa, Iowa City, Iowa, United States of America,
2 University of Iowa Carver College of Medicine, Iowa City, Iowa, United States of America, 3 Ilam University of Medical Sciences, Ilam, Iran, 4 Department of Neuroscience, Iran University of Medical Sciences, Tehran, Iran, 5 Community Family Medicine, Dartmouth Health, Lebanon, New Hampshire, United States of America, 6 Department of Epidemiology, University of Iowa, Iowa City, Iowa, United States of America

‡ FS and WRP contributed equally to this work and share first authorship. LGS and TLW contributed equally to this work and senior authorship.
* Terry-Wahls@uiowa.edu (TLW); Farnoosh-Shemirani@uiowa.edu (FS)

## Abstract

### Background

Multiple sclerosis (MS) is a chronic neuroinflammatory disease of the central nervous system, characterized by demyelination and neurodegeneration, which has a profound impact on the quality of life. Dysregulated inflammatory processes are a major driver of MS progression, with many areas of research being dedicated to modulating inflammation in people with MS. Several dietary patterns have been associated with improvements in inflammatory biomarkers; although, the findings have been inconsistent. Thus, this study aims to evaluate the effects of dietary interventions on inflammatory markers in adults with MS.

### Methods

Electronic databases, including PubMed/MEDLINE, Web of Science, Scopus, and Cochrane/Central, will be searched. Screening, selection, and extraction of data, along with quality assessment of included studies, will be done by two separate reviewers, and any potential conflicts will be settled through discussion. Two reviewers will independently assess the risk of bias in included studies using the Cochrane Risk of Bias Tool. If plausible, the results will be synthesized and pooled for meta-analysis. The overall quality of evidence of each study will be evaluated using the NutriGRADE tool, which is a modification to the Grading Recommendations Assessment, Development, and Evaluation (GRADE) developed specifically for nutrition research.

**Data Availability Statement:** The current study protocol does not report any data and the data availability policy is not applicable to this article. However, the search syntax utilized in the PubMed database has been provided as Supporting Information to facilitate the replication of the study findings. Deidentified research data will be made publicly available when the study is completed and published.

**Funding:** The author(s) disclosed they received financial support as follows: TJT, SMS, and FS are supported by the Carter Chapman Shreve Family Foundation and the Carter Chapman Shreve Fellowship Fund for diet and lifestyle research conducted by the Wahls Research team at the University of Iowa. TJT is additionally supported by a career transition award from the National MS Society (TA-2205-39486). The funders had no role in study design, data collection and analysis, decision to publish, or preparation of the manuscript. The authors received no specific funding for this work.

**Competing interests:** I have read the journal's policy and the authors of this manuscript have the following competing interests: Dr. Terry Wahls personally follows and promotes the Wahls™ diet. She has equity interest in the following companies: Terry Wahls LLC; TZ Press LLC; The Wahls Institute, PLC; FBB Biomed Inc; Levels Health Inc., Foogal Inc. and the website http://www.terrywahls. com. She also owns the copyright to the books Minding My Mitochondria (2nd Edition) and The Wahls Protocol, The Wahls Protocol Cooking for Life, and the trademarks The Wahls Protocol® and Wahls™ diet, Wahls Paleo™ diet, and Wahls Paleo Plus™ diets. She has completed grant funding from the National Multiple Sclerosis Society for the Dietary Approaches to Treating Multiple Sclerosis Related Fatigue Study. She has financial relationships with BioCeuticals Ltd., MCG Health LLC, Vibrant America LLC, Standard Process Inc., MasterHealth Technologies Inc., Foogal Inc., Genova Diagnostics Inc., and the Institute for Functional Medicine. She receives royalty payments from Penguin Random House. Dr Wahls has conflict of interest management plans in place with the University of Iowa and the Iowa City Veteran's Affairs Medical Center. All other authors report no personal or financial conflicts of interest in this work. This does not alter our adherence to PLOS ONE policies on sharing data and materials.

## Discussion

Studies have demonstrated conflicting results regarding the effects of dietary interventions on serum levels of inflammatory biomarkers among people with MS. Thus, it is expected that the planned systematic review and meta-analysis will yield robust evidence on the effects of diet on inflammatory profile in the setting of MS.

## 1. Introduction

Multiple sclerosis (MS) is a chronic neuroinflammatory disease with highly heterogeneous outcomes currently affecting approximately 2.8 million individuals globally [1]. There has been a significant rise in its prevalence, with a 30% global increase over the past decade. MS disproportionately affects women, with almost a two-fold greater incidence rate between genders, and is a frequent cause of permanent disability in young adults [2]. Existing evidence indicates the involvement of diverse risk factors in the development of MS, including genetic predispositions, exposure to infections, and elements contributing to pro-inflammatory conditions such as smoking, obesity, and insufficient sun exposure [3]. While significant progress has been made in unraveling the pathophysiology of MS, the exact underlying mechanisms remain elusive [4].

Much of the central nervous system (CNS) damage is believed to result from an immune-mediated processes, involving components of the innate immune system, as well as adaptive immune system activation of certain lymphocyte populations and B cell activation [5]. Some T cell populations, including T-helper (Th1) and Th17 lymphocytes, are associated with a repertoire of pro-inflammatory cytokines, while Th2 lymphocytes and T regulatory (Treg) cells secrete anti-inflammatory cytokines [6]. In people with MS, there is an imbalance between Th1/Th17 and T regulatory subset, resulting in increased Th1 pro-inflammatory cytokines such as interleukin-1 (IL-1), IL-6, and tumor necrosis factor (TNF-$\alpha$). In addition, secretion of Th2 anti-inflammatory cytokines, such as IL-10, is decreased in MS, and this can intensify the destruction of myelin and prevent remyelination [7]. In particular, CD4$^+$ T cells contribute to the initiation of the immune response in MS patients, while CD8$^+$ T cells, are believed to be the major lymphocytes found in active MS lesions, which have been identified as potential major contributors to MS pathology [8]. Tregs, as key regulators of the immune system, possess the ability to suppress autoimmunity and play a role in promoting tissue regeneration [8].

In addition to the immune-mediated mechanisms, compelling evidence suggests that immune-independent mediators contribute to the pathogenesis of MS [8]. In this regard, adipokines, which are white adipose tissue secretions, are believed to play a key role as immune cell modulators through altering the local and systemic inflammation [9]. Conversely, the function of adipokines is influenced by inflammatory factors released by immune cells. One example is leptin, an adipokine known to contribute to inflammatory mechanisms and neuro-degeneration in MS [10]. Additionally, it impacts immune system by regulating the balance between Th1 and Th2 [11]. While current disease-modifying treatments (DMTs) are effective for reducing relapses, disability progression, and lesions [12], they may not optimally modulate the complex balance of pro-inflammatory and protective immune responses [13, 14]. This finding, coupled with the reciprocal interplay between adipokines and the immune system, implies that additional adjunct approaches, including diet [15], may further optimize patient outcomes through modulation of cytokine and adipokine balance in MS.

Diet exerts systemic effects on human metabolism and is involved in immune system modulation and cytokine signaling, rendering it an attractive candidate as a potential

environmental mediator in MS [15]. Dietary factors that are able to promote regulatory, as opposed to inflammatory immune cell differentiation and cytokine production hold the potential to have beneficial effects on inflammation in MS [16]. Furthermore, evidence also supports the notion that there are links between specific foods and autoimmune diseases, namely MS [17], and that several specific dietary components are linked to inflammation [18–20]. Accordingly, nutrition research has shifted towards examining the effects of composite dietary patterns as opposed to focusing on individual nutrients [16]. Despite a lack of consistent evidence for any specific therapeutic diet for MS, preliminary trials suggest several dietary patterns may potentially reduce inflammation [18, 21, 22]; however, the specific effect of dietary patterns on inflammatory biomarkers among people with MS is an area with present insufficient amount of data and inconsistency. For example, a randomized control trial showed that adherence to a Mediterranean-style diet significantly decreases inflammatory markers as well as MS fatigue severity [23]. In contrast, Yadav et al. showed that a very-low fat plant-based diet is well tolerated by patients with MS, but had no significant improvement on inflammatory markers, namely hs-CRP [24].

Extensive research has thoroughly investigated the significant link between inflammation and its contribution to MS pathophysiology [25]. The ability of natural food components to modulate cell metabolism, specifically inflammatory and autoimmune processes, makes diet an intriguing prospect for MS therapy [19]. However, the specific effect of dietary patterns, rather than specific dietary components on inflammatory markers in MS, is an area with present inconsistency and an insufficient amount of data, and a systematic review (and meta-analysis) is crucial to consolidate the existing evidence. Exploring the link between dietary interventions and inflammation within the MS community is crucial for providing well-informed, evidence-based dietary recommendations to address a significant gap in current MS management practices. To the best of our knowledge, the current study will be the first to review and combine evidence on how diet affects inflammatory markers in people with MS.

## 2. Materials and methods

### 2.1. Study registration

The systematic review and meta-analysis (if possible) of dietary intervention trials was registered in PROSPERO International Prospective Register of Systematic Reviews (registration number: CRD42023425961). The current protocol was planned and developed according to the Preferred reporting items for systematic review and meta-analysis protocols (PRISMA-P) (S1 Checklist). Amendments to this protocol will be included in the updated versions of the PROSPERO, if needed. The systematic review and meta-analysis will be conducted according to the PRISMA 2020 guidelines [26] and following the Cochrane Collaboration Handbook for Systematic Reviews of Interventions [27].

### 2.2. Eligibility criteria

The justification for the eligibility criteria was conducted employing the patient, intervention, comparison, outcome, and study design (PICOs) framework. Studies that fulfill the following criteria will be selected: (1) studies that were published, in full, in a peer-reviewed journal; (2) studies that had the primary or secondary objective of assessing changes in inflammatory markers following any type of dietary intervention; (3) randomized trials; (4) studies conducted on participants of above 18 years of age, including all genders.

Studies will be excluded if they are: 1) studies conducted on animal models and/or cell culture; (2) studies including pregnant/lactating women, children, and adolescents; (3) Interventions based on supplementation or single nutrients and/or food groups; (4) Studies including

co-intervention restricted to only one group; (5) studies examining postprandial inflammatory responses after an immediate intervention; (6) studies with no control group; (7) case studies, review articles, study protocol papers, and conference abstracts.

**2.2.1. Population.** We will consider the adult population (18 years or more) with a diagnosis of MS. For the current study, RCTs that enrolled participants diagnosed with definitive MS according to McDonald criteria [28–30] will be included. No restrictions are to be placed on sub-types of MS.

**2.2.2. Intervention.** The intervention being studied will encompass any type of dietary pattern (defined as whole diets aimed at the maintenance or improvement of MS symptoms). Studies that evaluated single food items, nutrients, or meal replacement products (e.g., liquid shakes) will be excluded. The diet-based intervention needs to be described in sufficient detail, highlighting the main components of the intervention. No restrictions on the settings and type of intervention in terms of frequency, duration, type of diet, and delivery of intervention will be placed.

**2.2.3. Comparison (Control).** Comparisons will be made between the intervention diet vs. control group, such as standard or placebo diet, usual care, or any other dietary pattern described as the control.

**2.2.4. Outcome.** The primary outcome will be serum/plasma levels of inflammatory and immune-related biomarkers such as cytokines and adipocytokines. Studies measuring gene expression of inflammatory biomarkers will be excluded.

**2.2.5. Study design.** Randomized Controlled trials (RCTs), as the most reliable source of evidence for evaluation of health interventions, will be included in this study.

## 2.3. Search strategy

We will search the following electronic databases: PubMed/MEDLINE, Cochrane Central Register of Controlled Trials (CENTRAL), Scopus, and Web of Science. The search strategy was developed in collaboration with a medical librarian with expertise in systematic reviews. The search strategy will include MESH Terms and keywords related to the intervention, health concern, and outcome(s) of interest. The search syntax for PubMed/Medline database can be found in S1 File. The search terms will be adapted for use with other databases in combination with database-specific filters. No restrictions will be imposed for language or date of publication. After retrieval of full-text articles, the reference lists of the articles and other related reviews on dietary patterns and inflammation will be hand-searched to identify additional potentially relevant articles that could be included. Gray literature, such as conference papers as well as unpublished records, will not be included in the meta-analysis. In order to remove duplicated studies and merge retrieved citations, EndNote (version 20) will be used to facilitate the screening processes. The search will be rerun prior to the final analysis.

## 2.4. Data collection and analysis

**2.4.1. Study selection.** References from various databases and repositories will be imported into EndNote20 referencing software. Two reviewers will independently review the titles, abstracts, and full-text papers of the identified articles for inclusion, and any duplicates will be removed using EndNote. All searches will follow a two-phase screening strategy:

1. Phase one will assess the eligibility of the title and abstract of every manuscript generated from the electronic and hand-searching against the predetermined inclusion and exclusion criteria. Studies that either do not clearly meet the inclusion criteria or meet at least one exclusion criteria will be excluded at this phase.

2. Once the first round of screening is completed, the full-text paper will be retrieved from articles identified in stage one and will be assessed against the predetermined inclusion and exclusion criteria.

Any conflicts between reviewers relating to study eligibility will be resolved in consensus meetings with a third and fourth reviewer. The results of searches in all databases, along with the selection process, will be displayed using the PRISMA 2020 flow diagram (Fig 1).

**2.4.2. Data extraction.** Data extraction will be conducted by one reviewer using a pre-piloted data extraction form and independently verified by a different review team member. Study characteristics (i.e., author's name, year of publication, country, study design), population characteristics (i.e., number of participants, number of participants in the final analysis, age, gender, intervention duration, and type, co-interventions), and outcomes (inflammatory

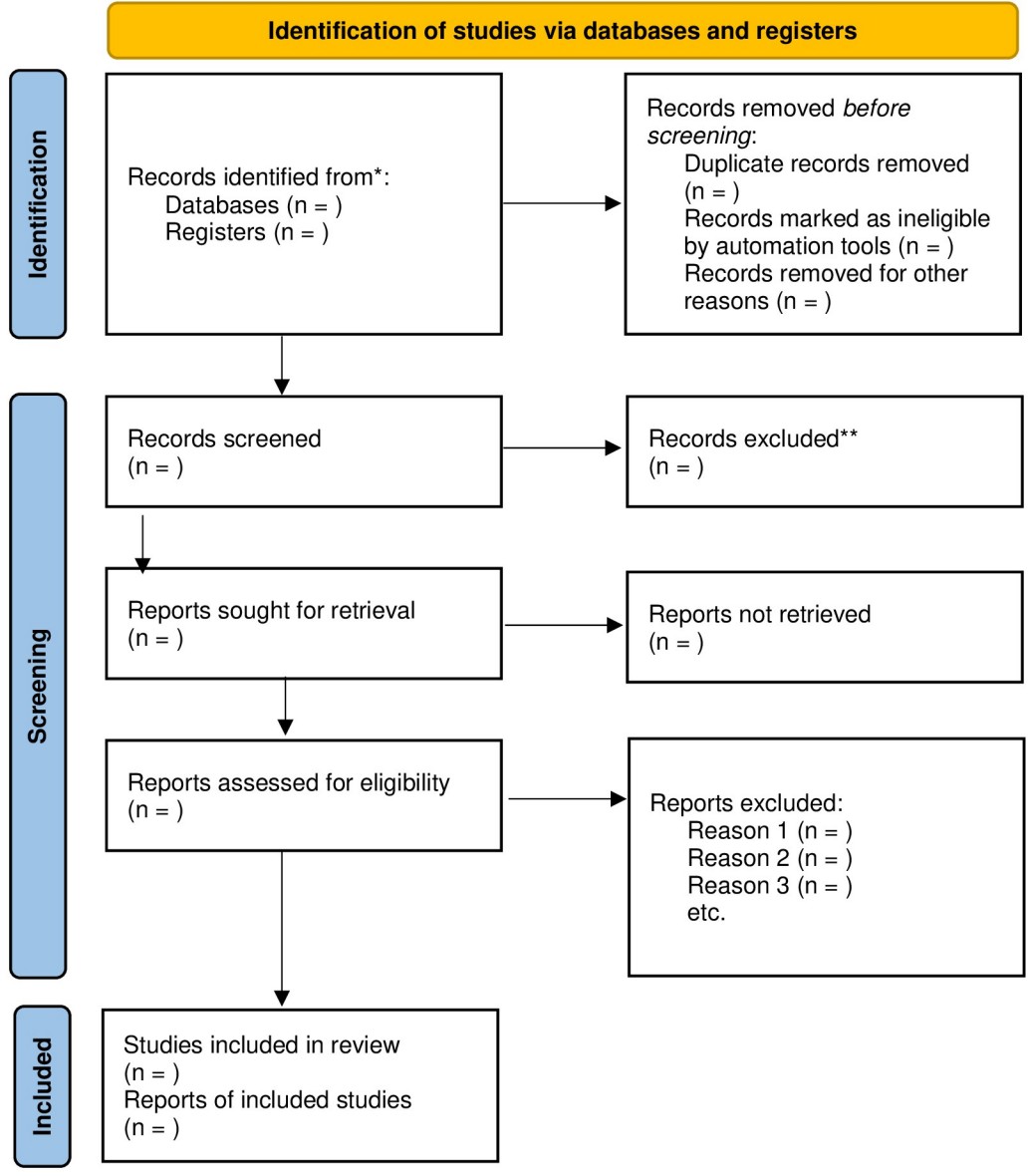

**Fig 1. PRISMA flow diagram for identifying, screening, and determining the eligibility and inclusion of studies.**

biomarkers/ unit of assessment e.g., measurement tool or method used, results of interventions e.g., mean difference, confidence intervals, P-values, etc.) will be extracted. Any discrepancies will be achieved by discussion during consensus meetings, or, if needed, in consultation with a third and fourth reviewer. The information will be summarized in a "table of characteristics" (S1 Table).

**2.4.3. Risk-of-bias analysis.**   Two reviewers will independently assess the risk of bias in included studies using the Cochrane Risk of Bias Tool [31]. Risk of bias will be judged as low, high, or unclear for individual elements: 1) random sequence generation (selection bias); 2) allocation concealment (selection bias); 3) blinding of outcome assessment (detection bias) and self-reported outcomes; 4) incomplete outcome data addressed (attrition bias); 5) selective reporting (reporting bias); 6) other bias (other sources of bias that have been detected by the reviewer). Two figures will be generated to demonstrate the risk of bias assessment of individual and overall studies that have been included in the systematic review and meta-analysis.

**2.4.4. Credibility of evidence.**   The credibility of evidence for each pairwise comparison will be evaluated using the NutriGRADE scoring system [32], which is an extension of the Grading of Recommendations Assessment, Development, and Evaluation (GRADE) scoring system that is specific to nutrition studies and comprises the following items: (1) risk of bias, study quality, and study limitations; (2) precision; (3) heterogeneity; (4) directness; (5) publication bias; (6) funding bias; and (7) study design for meta-analyses of randomized trials. This scoring system judges the meta-evidence of each pairwise comparison and categorizes scores into 4 classes: high, moderate, low, and very low credibility of evidence.

## 2.5. Statistical analysis

A random effects meta-analysis will be performed on randomized controlled trials. Stata software (version 12) will be employed to conduct all statistical procedures. The mean and SD of changes in inflammatory markers from baseline will be used to calculate the overall effect size and will be used to compare the effects of diets in both intervention and control groups. Besides visual inspection of the forest plots, the presence, and the extent of heterogeneity we assessed with the Cochrane's Q-test and I-squared ($I^2$) index, respectively. In order to consider heterogeneity, the random-effects model will be adopted to estimate standardized mean differences (SMD) with 95% confidence intervals (CIs). Potential publication bias will be examined through Eggers' regression symmetry test and visual examination of funnel plots. To evaluate possible sources of heterogeneity, subgroup analysis, and meta-regression analysis will be conducted (if there are a sufficient number of studies). Sensitivity analysis will be carried out to explore whether the overall effect of diets on inflammatory markers is dependent on a particular study. $P < 0.05$ will be set as statistically significant. The overall quality of evidence will be evaluated using the NutriGRADE tool.

## 3. Ethics and dissemination

Ethics committee approval and informed consent from patients will not be required. The planned systematic review and meta-analysis will carry public health significance by offering current evidence regarding the effect of dietary patterns on inflammatory biomarkers among adults with MS. The results of the systematic review and meta-analysis will be submitted to a peer-reviewed journal.

## 4. Discussion

The objective of this study will be to evaluate the impact of dietary interventions on inflammatory biomarkers among individuals with MS. Multiple studies have consistently pointed

toward the significance of the inflammatory processes in driving the pathogenesis and exacerbation of MS [8, 33]. The ability of dietary patterns to improve inflammatory and autoimmune processes makes diet an intriguing prospect for individuals with MS as a potential adjunct therapy [19]. To our knowledge, this systematic review and meta-analysis will be the first to summarize and compare the effects of different dietary approaches on inflammatory markers in people with MS. The findings will add more evidence to the literature on the potential effects of dietary interventions on the inflammatory and immune-regulating biomarkers in the setting of MS. Given improving the inflammatory process and shifting the balance towards anti-inflammatory biomarkers is important for ameliorating the MS symptoms and specific dietary recommendations for individuals with MS are scarce, this systematic review will provide important data regarding the role of diet in inflammatory pathways in MS.

## Supporting information

**S1 Checklist. PRISMA-P checklist.**
(PDF)

**S1 File. Search syntax for PubMed database.**
(PDF)

**S1 Table. Characteristics of included studies in the systematic review and meta-analysis.**
(PDF)

## Acknowledgments

The authors would like to acknowledge the following contributor: Heather Healy, a health sciences librarian, for her valuable assistance in developing the search syntax.

## Author Contributions

**Investigation:** Wade R. Pingel.

**Methodology:** Farnoosh Shemirani, Farshad Arsalandeh.

**Supervision:** Linda G. Snetselaar, Terry L. Wahls.

**Writing – original draft:** Farnoosh Shemirani, Wade R. Pingel.

**Writing – review & editing:** Tyler J. Titcomb, Asma Salari-Moghaddam, Farshad Arsalandeh, Solange M. Saxby, Linda G. Snetselaar, Terry L. Wahls.

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
