## [Decision Letter · Decision Letter 0]

27 Oct 2023

PONE-D-23-29197The Effect of Dietary Interventions on Inflammatory Biomarkers Among People with Multiple Sclerosis: A Protocol for Systematic Review and Meta-Analysis of Randomized Controlled Trials

PLOS ONE

Dear Dr. Wahls,

Thank you for submitting your manuscript to PLOS ONE. After careful consideration, we feel that it has merit but does not fully meet PLOS ONE’s publication criteria as it currently stands. Therefore, we invite you to submit a revised version of the manuscript that addresses the points raised during the review process.

Please note that I have acted as a reviewer for this manuscript, and you will find my comments below, under Reviewer 3.

Furthermore, we note that Reviewer 1 has recommended that you cite specific previously published works. As always, we recommend that you please review and evaluate the requested works to determine whether they are relevant and should be cited. It is not a requirement to cite these works. We appreciate your attention to this request.

Based on the reviewers' suggestions, the paper needs major revision.  The reviewers' comments can be found below.

We look forward to receiving your revised manuscript.

Kind regards,

Tanja Grubić Kezele, Ph.D., M.D.

Academic Editor

PLOS ONE

Journal Requirements:

https://www.sciencedirect.com/science/article/abs/pii/S1043466621001733?via%3Dihub

https://core.ac.uk/reader/227578950

In your revision ensure you cite all your sources (including your own works), and quote or rephrase any duplicated text outside the methods section. Further consideration is dependent on these concerns being addressed.

3. Thank you for stating the following financial disclosure: ""N/A""

4. Thank you for stating the following in the Competing Interests section: ""I have read the journal's policy and the authors of this manuscript have the following competing interests: 

Dr. Terry Wahls personally follows and promotes the Wahls™ diet. She has equity interest in the following companies: Terry Wahls LLC; TZ Press LLC; The Wahls Institute, PLC; FBB Biomed Inc; Levels Health Inc., Foogal Inc. and the website http://www.terrywahls.com. She also owns the copyright to the books Minding My Mitochondria (2nd Edition) and The Wahls Protocol, The Wahls Protocol Cooking for Life, and the trademarks The Wahls Protocol® and Wahls™ diet, Wahls Paleo™ diet, and Wahls Paleo Plus™ diets. She has completed grant funding from the National Multiple Sclerosis Society for the Dietary Approaches to Treating Multiple Sclerosis Related Fatigue Study. She has financial relationships with BioCeuticals Ltd., MCG Health LLC, Vibrant America LLC, Standard Process Inc., MasterHealth Technologies Inc., Foogal Inc., Genova Diagnostics Inc., and the Institute for Functional Medicine. She receives royalty payments from Penguin Random House. Dr Wahls has conflict of interest management plans in place with the University of Iowa and the Iowa City Veteran’s Affairs Medical Center. All other authors report no personal or financial conflicts of interest in this work.""

Please confirm that this does not alter your adherence to all PLOS ONE policies on sharing data and materials, by including the following statement: ""This does not alter our adherence to  PLOS ONE policies on sharing data and materials."" (as detailed online in our guide for authors http://journals.plos.org/plosone/s/competing-interests).  If there are restrictions on sharing of data and/or materials, please state these. Please note that we cannot proceed with consideration of your article until this information has been declared. 

We will update your Data Availability statement on your behalf to reflect the information you provide."

Reviewers' comments:

Reviewer's Responses to Questions

**Comments to the Author**

1. Does the manuscript provide a valid rationale for the proposed study, with clearly identified and justified research questions?

Reviewer #1: Yes

Reviewer #2: Yes

Reviewer #3: Partly

2. Is the protocol technically sound and planned in a manner that will lead to a meaningful outcome and allow testing the stated hypotheses?

Reviewer #1: Yes

Reviewer #2: Yes

Reviewer #3: Yes

3. Is the methodology feasible and described in sufficient detail to allow the work to be replicable?

Reviewer #1: No

Reviewer #2: Yes

Reviewer #3: Yes

4. Have the authors described where all data underlying the findings will be made available when the study is complete?

Reviewer #1: Yes

Reviewer #2: Yes

Reviewer #3: Yes

5. Is the manuscript presented in an intelligible fashion and written in standard English?

Reviewer #1: Yes

Reviewer #2: Yes

Reviewer #3: Yes

6. Review Comments to the Author

You may also provide optional suggestions and comments to authors that they might find helpful in planning their study.

Reviewer #1: The study would benefit from a special trigger on known cross-reactivity of myelin-related proteins and dietary proteins.

Inclusion of the following references in the introduction may also benefit the work for comprehensiveness:

https://doi.org/10.1093/intimm/dxaa054

https://doi.org/10.1016/j.jtauto.2019.100003

https://doi.org/10.1016/j.autrev.2020.102459

Reviewer #2: The Effect of Dietary Interventions on Inflammatory Biomarkers Among People with Multiple Sclerosis: A Protocol for Systematic Review and Meta-Analysis of Randomized Controlled Trials

The article/protocol is very good and useful for specialists who want to perform the systematic review and meta-analysis of randomized controlled trials regarding the effect of dietary interventions on inflammatory biomarkers among people with

multiple sclerosis.

In addition, systematic reviews and meta-analyses of clinical trials concerning these aspects in MS are lacking. Thus, I agree with the publication of this study protocol. Moreover, the manuscript is well-structured and comprehensible.

Reviewer #3: The authors aim to evaluate the effects of dietary interventions

on inflammatory markers in adults with MS.

Comments:

Introduction: lines 56-57: "The induction of inflammation by the immune system is linked with several factors, including

adipose tissue-associated cytokine and adipokine production (7, 8)."

From this sentence it seems like adipokines are the main mechanism involved in the MS pathophysiology. The induction of inflammation could be genetically provoked in genetically susceptible individuals, by microorganisms, by hypovitaminoses D, etc. Please insert more explanations and additional references.

7. PLOS authors have the option to publish the peer review history of their article (what does this mean?). If published, this will include your full peer review and any attached files.

Reviewer #1: No

Reviewer #2: **Yes: **Lecturer Camelia Munteanu

Reviewer #3: No

---

## [Author Response · Author response to Decision Letter 0]

5 Dec 2023

Responses to Reviewers' comments

The authors express their gratitude to all the reviewers for providing valuable comments and dedicating efforts to enhance our manuscript. Diligent attention has been given to refining the work and addressing their concerns. All requested journal requirements have been addressed and included either in the revised manuscript or within the cover letter. The following are the point-by-point responses to the reviewers’ comments/questions.

Reviewers' comments:

Reviewer's Responses to Questions

Comments to the Author

1. Does the manuscript provide a valid rationale for the proposed study, with clearly identified and justified research questions?

Reviewer #1: Yes

Reviewer #2: Yes

Reviewer #3: Partly

2. Is the protocol technically sound and planned in a manner that will lead to a meaningful outcome and allow testing the stated hypotheses?

Reviewer #1: Yes

Reviewer #2: Yes

Reviewer #3: Yes

3. Is the methodology feasible and described in sufficient detail to allow the work to be replicable?

Reviewer #1: No

Reviewer #2: Yes

Reviewer #3: Yes

4. Have the authors described where all data underlying the findings will be made available when the study is complete?

Reviewer #1: Yes

Reviewer #2: Yes

Reviewer #3: Yes

5. Is the manuscript presented in an intelligible fashion and written in standard English?

Reviewer #1: Yes

Reviewer #2: Yes

Reviewer #3: Yes

Reviewer #1: 

Q1: The study would benefit from a special trigger on known cross-reactivity of myelin-related proteins and dietary proteins.

A1: We appreciate the valuable suggestion from the reviewer. Exploring the interplay between myelin-related proteins and dietary proteins is an intriguing topic that could be considered for investigation in a separate systematic review. The present protocol for systematic review and meta-analysis is focused on examining the potential impacts of dietary patterns, as opposed to individual dietary components, on alterations in serum/plasma levels of inflammatory biomarkers, including adipokines and cytokines.

Q2: Inclusion of the following references in the introduction may also benefit the work for comprehensiveness:

https://doi.org/10.1093/intimm/dxaa054

https://doi.org/10.1016/j.jtauto.2019.100003

https://doi.org/10.1016/j.autrev.2020.102459

A2: We appreciate the reviewer for offering the inclusion of the suggested references. The three references primarily delve into the dietary antigens of specific food components such as pork, milk, and cheese. They explore the potential correlation between these components and autoimmunity, discussing topics like dietary-human autoimmune epitope overlap, immunogenetics, and the potential autoimmunogenicity of certain foods. To enhance the comprehensiveness of the manuscript, we have incorporated the second reference (https://doi.org/10.1016/j.jtauto.2019.100003) into the introduction section (line 84, reference 18).

Q3. Is the methodology feasible and described in sufficient detail to allow the work to be replicable?

Reviewer #1: No

A3: In addressing the reviewer's concern about the feasibility and clarity of the methodology for potential replication, we want to highlight that the search syntax employed in the PubMed database is available in S2 Table, aiding in the replication of our study's findings. Furthermore, a comprehensive methodological description, encompassing eligibility criteria, study design, search strategy, study selection, data extraction, and credibility assessment of evidence, can be found in Section 2 and its subheadings (2.1 – 2.5).

Reviewer #2: 

The Effect of Dietary Interventions on Inflammatory Biomarkers Among People with Multiple Sclerosis: A Protocol for Systematic Review and Meta-Analysis of Randomized Controlled Trials.

The article/protocol is very good and useful for specialists who want to perform the systematic review and meta-analysis of randomized controlled trials regarding the effect of dietary interventions on inflammatory biomarkers among people with multiple sclerosis.

In addition, systematic reviews and meta-analyses of clinical trials concerning these aspects in MS are lacking. Thus, I agree with the publication of this study protocol. Moreover, the manuscript is well-structured and comprehensible.

A3: We appreciate the reviewer's positive feedback on our protocol. We are grateful for their recognition of the manuscript's structure and clarity, as well as for supporting the publication of our study protocol.

Reviewer #3: 

The authors aim to evaluate the effects of dietary interventions on inflammatory markers in adults with MS.

Comments:

Introduction: lines 56-57: "The induction of inflammation by the immune system is linked with several factors, including adipose tissue-associated cytokine and adipokine production (7, 8)."

Q4: From this sentence it seems like adipokines are the main mechanism involved in the MS pathophysiology. The induction of inflammation could be genetically provoked in genetically susceptible individuals, by microorganisms, by hypovitaminoses D, etc. Please insert more explanations and additional references.

A4: We thank the reviewer for making this comment. We have added additional explanation and cited more relevant references for more clarity. Please see lines 47-50, and 66-73 in the revised manuscript. 

Q5: If applicable, we recommend that you deposit your laboratory protocols in protocols.io to enhance the reproducibility of your results. Protocols.io assigns your protocol its own identifier (DOI) so that it can be cited independently in the future. For instructions see: https://journals.plos.org/plosone/s/submission-guidelines#loc-laboratory-protocols. Additionally, PLOS ONE offers an option for publishing peer-reviewed Lab Protocol articles, which describe protocols hosted on protocols.io. Read more information on sharing protocols at https://plos.org/protocols?utm_medium=editorial-email&utm_source=authorletters&utm_campaign=protocols.

A5: Thank you for your suggestion. It's important to note that the present manuscript serves as a protocol for a systematic review and meta-analysis, categorized as a Study Protocol. We have adhered to the reporting guidelines specific to protocols, namely PRISMA-P, as outlined in the "Guidelines for Specific Study Types", and have already registered the study protocol in PROSPERO International Prospective Register of Systematic Reviews (registration number: CRD42023425961). We will make sure to choose the correct study type when submitting the revision.

Q6: Does the manuscript provide a valid rationale for the proposed study, with clearly identified and justified research questions?

Reviewer #3: Partly

A6: We appreciate the reviewer for raising this concern. in response, we would like to highlight the notion that our research rationale stems from the gaps observed in the current literature. Clinically, the course of MS has been associated with inflammatory mediators including cytokines such as IL-1β, IL-6, IL-17, TNF-α, and INF-γ. The extensive body of research on inflammation in MS has laid a foundation, pointing towards the potential therapeutic role of diet. Because many pharmacological treatments have significant limitations, individuals with MS seek non-pharmacologic therapies to mitigate fatigue and reduce their symptoms. Clinical trials and animal studies support that diet profoundly impacts MS severity and disease trajectory. Yet, evidence for the mechanism of action of therapeutic diets, as opposed to individual dietary components, remains inconsistent. 

Given this inconsistency and the limited role of neurologists in providing dietary recommendations, newly diagnosed people with MS typically receive minimal dietary guidance. In turn, many individuals seek dietary information from internet sources, which are not often evidence-based. Thus, the specific influence of dietary patterns on inflammatory markers lacks comprehensive exploration, and a systematic review is crucial to consolidate the existing evidence. 

Providing further evidence on the link between dietary interventions and inflammation within the MS community stands to benefit not only individuals dealing with MS but also their healthcare providers, support persons, and families. It's crucial to provide well-informed, evidence-based dietary guidance to address a significant gap in current MS management practices. The last paragraph of the introduction (lines 95-105) has been revised to articulate the study's rationale more clearly.

---

## [Decision Letter · Decision Letter 1]

8 Jan 2024

The Effect of Dietary Interventions on Inflammatory Biomarkers Among People with Multiple Sclerosis: A Protocol for Systematic Review and Meta-Analysis of Randomized Controlled Trials

PONE-D-23-29197R1

Dear Dr. Wahls,

We’re pleased to inform you that your manuscript has been judged scientifically suitable for publication and will be formally accepted for publication once it meets all outstanding technical requirements.

Kind regards,

Tanja Grubić Kezele, Ph.D., M.D.

Academic Editor

PLOS ONE

Additional Editor Comments (optional):

Reviewers' comments:

Reviewer's Responses to Questions

**Comments to the Author**

1. Does the manuscript provide a valid rationale for the proposed study, with clearly identified and justified research questions?

Reviewer #2: Yes

Reviewer #3: Yes

2. Is the protocol technically sound and planned in a manner that will lead to a meaningful outcome and allow testing the stated hypotheses?

Reviewer #2: Yes

Reviewer #3: Yes

3. Is the methodology feasible and described in sufficient detail to allow the work to be replicable?

Reviewer #2: Yes

Reviewer #3: Yes

4. Have the authors described where all data underlying the findings will be made available when the study is complete?

Reviewer #2: Yes

Reviewer #3: Yes

5. Is the manuscript presented in an intelligible fashion and written in standard English?

Reviewer #2: Yes

Reviewer #3: Yes

6. Review Comments to the Author

You may also provide optional suggestions and comments to authors that they might find helpful in planning their study.

Reviewer #2: As a researcher, I found this study protocol very useful. As I wrote the first time, I agree with its publication. Congratulations to all co-authors!

Reviewer #3: The authors have solved all queries.

The paper can be published.

The introduction, methods part, discussion part are well described.

7. PLOS authors have the option to publish the peer review history of their article (what does this mean?). If published, this will include your full peer review and any attached files.

Reviewer #2: No

Reviewer #3: No

---

## [Editor Report · Acceptance letter]

25 Jan 2024

PONE-D-23-29197R1 

PLOS ONE

Dear Dr. Wahls, 

I'm pleased to inform you that your manuscript has been deemed suitable for publication in PLOS ONE. Congratulations! Your manuscript is now being handed over to our production team.

Kind regards, 

on behalf of

Prof. dr. Tanja Grubić Kezele 

Academic Editor

PLOS ONE